# VaCRK2 Mediates Gray Mold Resistance in *Vitis amurensis* by Activating the Jasmonate Signaling Pathway

Tinggang Li [1], Huanhuan Gao [1], Xiaoning Tang [1] and Dongying Gong [2],*

1 Shandong Academy of Grape, Shandong Academy of Agricultural Sciences, No. 1–27, Shanda South Road, Jinan 250100, China; weifengluolu@126.com (T.L.); litinggang@yeah.net (H.G.); txn101@126.com (X.T.)
2 Institute of Agro-Food Sciences and Technology, Shandong Academy of Agricultural Sciences, No. 202, Gongye North Road, Jinan 250100, China
* Correspondence: putaoyuan@shandong.cn

**Abstract:** Cysteine-rich receptor-like kinases (CRKs) are ubiquitous plant receptor-like kinases, which play a significant role in plant disease resistance. Gray mold is an economically important disease of grapes caused by *Botrytis cinerea*. However, *CRK* genes and their function in gray mold disease resistance in grapes have not been elucidated. This study aimed to identify and characterize *CRKs* in grapes and determine their role in gray mold resistance. Four *CRKs* were identified in *Vitis amurensis* and named *VaCRK1–VaCRK4* according to their genomic distribution. The four *VaCRKs* were ectopically expressed in *Arabidopsis thaliana* to study their function in defense response against *B. cinerea*. Heterologous expression of *VaCRK2* in *A. thaliana* conferred resistance to *B. cinerea*. *VaCRK2* expression in gray mold-resistant grape cultivar increased significantly after *B. cinerea* inoculation and methyl jasmonate treatment. Furthermore, the expression of jasmonic acid (JA) signaling pathway-related genes in *VaCRK2* overexpression lines of *A. thaliana* was significantly increased after *B. cinerea* inoculation, leading to the upregulation of pathogenesis-related (PR) genes and reactive oxygen species (ROS) accumulation. Overall, these results suggest that *VaCRK2* confers resistance to *B. cinerea* by activating PR gene expression and oxidative burst through the JA signaling pathway.

**Keywords:** grape; gray mold; jasmonic acid; reactive oxygen species; pathogenesis-related gene

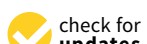

## 1. Introduction

Two layers of immune system response have evolved during the arms race between plants and phytopathogens, including bacteria, fungi and viruses [1]. One involves the recognition of specific molecular motifs conserved within a class of pathogens, also known as pathogen-associated molecular patterns (PAMP), through receptors located on the plant cell membrane, thereby activating the PAMP-triggered immunity (PTI) to resist pathogen invasion [2]. However, sometimes pathogens manage to bypass PTI by secreting effectors into the plant cell. In response, plants evolved another form of immune response involving resistance (R) proteins. These proteins perceive specific virulence effectors secreted by pathogens, thereby triggering the effector-triggered immunity (ETI), a more robust immune response against pathogen attack [3]. Recent studies have found that ETI can increase the expression of core PTI components, thereby amplifying the PTI and inducing a durable immune response [4]. For example, ETI enhances the expression of respiratory burst oxidase homolog D (RBOHD) protein, while PTI promotes the complete activation of RBOHD protein. This cooperation mechanism ensures that plants mount a rapid and precise response against pathogen infection while maintaining optimal growth and development [4].

PTI response is activated when receptors on the plant cell membrane interact with specific pathogen molecular patterns. Plants use a variety of plasma membrane-bound receptor proteins to sense exogenous immune signals [5]. These cell surface receptor proteins

have different extracellular domains for sensing specific ligands. Receptor-like kinases (RLKs) are an important type of receptor proteins mainly composed of an N-terminal signal peptide, a variable extracellular domain, a transmembrane domain and a C-terminal intracellular kinase domain [6]. They can be divided into different subfamilies based on the nature of their variable extracellular domain, including leucine-rich repeats (LRRs), lectins (Lec), lysine motifs (LysMs) and cell wall-related kinases (WAKs) [7]. Generally, the N-terminus of RLKs, including signal peptide and variable extracellular domain, extends to the apoplast and can sense the signals released by pathogens. Meanwhile, the C-terminal kinase domain is located in the cytoplasm and transmits signals into the intracellular space to activate the plant cell immune response [8].

Cysteine-rich receptor-like kinases (CRKs) belong to an important class of RLKs that plays a vital function in plant disease resistance [9]. The extracellular domain of CRKs comprises two DUF26 motifs and a conserved C-X8-C-X2-C motif. The conserved Cys residues of the C-X8-C-X2-C motif facilitate the formation of the three-dimensional structure of proteins through disulfide bonds or zinc finger motifs, like in many DNA binding transcription factors [6]. Notably, disulfide bond and zinc finger can mediate protein–protein interaction, which is a key step in receptor activation [6]. The role of *CRKs* in plant defense response has been revealed in many plants [10–14]. In *Arabidopsis thaliana*, overexpression of *CRK13* and *CRK28* improves resistance to *Pseudomonas syringae* [10,11]. In wheat, *TaCRK2* enhances resistance to *Puccinia triticina* and has a positive effect on pathogen-induced plant cell death [12]. Rice *CRK10* mediates the resistance of rice to *Xanthomonas oryzae* pv. *oryzae* by interacting with transcription factor TGA2.1 [13]. In addition, *HvCRK1* negatively regulates the basal resistance of barley to *Blumeria graminis* f. sp. *hordei* [14].

Gray mold disease is mainly caused by *Botrytis cinerea*, a common fungal pathogen infecting more than 1400 plant species, including tomato, strawberry, grape and pepper [15]. Gray mold is the primary disease during the growth, storage and transportation of grapes [15]. It causes 20–50% yield losses in grapes if not controlled [16]. Improving host disease resistance is the most economical and effective strategy for controlling *B. cinerea*. However, the process requires identifying resistance genes, which can then be incorporated into the grape breeding programs. At present, a few genes that can confer resistance or tolerance to *B. cinerea* have been identified in grape, including *VvNPR1.1* [16], *VaSTS19* [17], *VlWRKY3* [18], *VqSTS21* [19] and *VqAP13* [20]. The introduction of *VaSTS19* and *VvNPR1.1* genes into *A. thaliana* improved the resistance to *B. cinerea* [16,17]. In contrast, overexpression of *VlWRKY3*, *VqSTS21,* and *VqAP13* genes in *A. thaliana* increased susceptibility to *B. cinerea* [18–20]. The demand for gray mold resistance genes in grape breeding has continued to increase with the continuous evolution of *B. cinerea*. Thus, there is a need to mine new resistance genes against the pathogen.

*Vitis amurensis* is a grape species that exhibits excellent resistance to diverse diseases, including gray mold. It is, therefore, an important source of disease resistance genes [21]. To the best of our knowledge, there is no report on the disease resistance function of *CRK* genes in grapes. This study (1) identified *CRK* genes in the *V. amurensis* genome and validated their role in *B. cinerea* resistance in *A. thaliana*, (2) analyzed the expression patterns of the candidate gene in response to *B. cinerea* infection and hormone induction, (3) evaluated the relationship between hormones and plant defense response mediated by candidate gene and (4) identified the signaling pathway of candidate gene-mediated resistance to *B. cinerea*. This study illuminates the role of the grape *CRK* gene in resistance against *B. cinerea* and provides a reference for molecular breeding of grapes for resistance against gray mold disease.

## 2. Materials and Methods

### 2.1. Identification of the VaCRKs

Genomic data of *V. amurensis* were download from http://www.grapeworld.cn/am/download/ (accessed on 11 December 2020). HMMER program was used to match Stress-antifung protein and Pkinase protein based on the HMM profile of Stress-antifung

(PF01657) domain and Pkinase (PF00069) domain in *V. amurensis* genome with an E-value cutoff of e$^{-5}$ [22]. SignalP-5.0 (http://www.grapeworld.cn/am/download/ (accessed on 15 December 2020)) and TMHMM Server v. 2.0 (http://www.cbs.dtu.dk/services/TMHMM/ (accessed on 15 December 2020)) were used to verify the presence of signal peptides and transmembrane domain, respectively [23,24]. Putative genes were further verified based on the SMART website and InterProScan database [25,26].

### 2.2. Analysis of Subcellular Localization of VaCRK2

The primers of *VaCRK2* gene were designed according to the full-length *VAG0104573.1* sequence. The full-length *VaCRK2* gene was amplified by PCR using cDNA of *V. amurensis* cultivar (cv. Shuangyou) as template. Purified PCR products were then inserted into pBin-*GFP4* vector, fused with *GFP* gene, under the control of cauliflower mosaic virus 35S promoter [6]. The vectors p35S:*GFP* and p35S:*VaCRK2* were transiently expressed in tobacco leaf epidermal cells using the *Agrobacterium tumefaciens* infiltration method [12]. Laser scanning confocal microscope (LSMT-PMT) was used to examine the subcellular localization of the proteins at 488 and 510 nm excitation and emission wavelengths, respectively.

### 2.3. Grape Plant Materials and Treatments

Grape species *V. amurensis* cv. Shuangyou and *V. vinifera* cv. Red Globe were used for the experiments. Notably, cv. Shuangyou is resistant to *B. cinerea*, whereas cv. Red Globe is susceptible. *B. cinerea* strain *BcSD3* (highly virulent strain) was cultured in potato dextrose agar (PDA) medium to produce conidia. Conidia were eluted with sterile deionized water and adjusted to a concentration of $2 \times 10^6$ conidia/mL. Grape leaves were inoculated with *BcSD3* and harvested at 0, 2, 6, 12, 24, 48, 72 and 120 h time points to assess resistance to *B. cinerea*. Grape leaves were sprayed with 10 mM methyl jasmonate (MeJA), 10 mM salicylic acid (SA), 10 mM ethephon (ETH) and 100 mM abscisic acid (ABA) to explore related signaling pathways. Leaves of control plants were sprayed with sterile distilled water. The leaves were collected at 0, 2, 6, 12, 24, 48 and 72 h time points. Six leaves per replicate, repeated three times, were harvested at each time point, frozen in liquid nitrogen and stored at −80 °C for subsequent analysis.

### 2.4. Generation and Analysis of Transgenic A. thaliana

The ORF sequence of *VaCRK2* gene was integrated into vector *pBI121* under the control of cauliflower mosaic virus 35S promoter. The vector was then transformed into *Agrobacterium tumefaciens* strain GV3101 using freeze–thaw method [18]. *VaCRK2* gene was introduced into *A. thaliana* wildtype Col-0 using *A. tumefaciens* mediated transformation [27]. Putatively transformed events were analyzed using PCR and RT-PCR to identify T$_3$ homozygous transgenic plants. Genomic DNA and cDNA from *A. thaliana* wildtype Col-0 were used as the control for PCR and RT-PCR detection [19].

### 2.5. Evaluation of Transgenic A. thaliana for Resistance against B. cinerea

*A. thaliana* wildtype Col-0 and *VaCRK2* overexpression lines (OE1, OE2 and OE3) were used for the experiments. *BcSD3* strain was cultured in PDA medium for five days. Conidia were eluted with sterile deionized water and adjusted to a concentration of $2 \times 10^6$ conidia/mL. Three-week-old *A. thaliana* plants were inoculated with *B. cinerea* conidia suspension. After inoculation, the plants were incubated at 22 °C under 16 h photoperiod with a humidity of 80–100%. A total of 24 leaves of six plants (four leaves per plant) were inoculated in one experiment, and the experiment was repeated thrice. The lesion diameter and fungal biomass were determined three days post-inoculation. Six leaves per replicate were harvested at each time point. The samples were frozen in liquid nitrogen and stored at −80 °C for subsequent analysis.

### 2.6. Oxidative Burst Assay in A. thaliana

*A. thaliana* wildtype Col-0 and *VaCRK2* overexpression 1 line (OE1) were used for the experiments. Reactive oxygen species accumulation was determined by diaminobenzidine (DAB) staining. Three-week-old *A. thaliana* leaves were infiltrated with 10 μL conidia suspension of *B. cinerea*. The *A. thaliana* leaves were infiltrated with sterile water as a control. After 12 h, the leaves were immersed in DAB buffer and vacuum infiltrated at 25 °C in the dark. The reaction was terminated after 12 h, and the leaves were washed with sterile water, bleached using 75% ethanol, and then immersed in 30% glycerin. ImageJ software was used to determine the percentage of brown pixels in the leaves. Antioxidant enzymes, including peroxidase (POD) and catalase (CAT), were extracted from *A. thaliana* leaves and measured as described by Guo et al. [18].

### 2.7. Relative Gene Expression Analysis

PLANTeasy kit (YPH-Bio, Beijing, China) was used to extract total RNA. After extraction and quality verification, the RNA samples were reverse transcribed to cDNA using TIANSeq M-MLV (RNase H-) reverse transcriptase (Tiangen, Beijing, China) according to the manufacturer's instructions. Then, RT-qPCR analysis was performed using the SYBR Premix Ex Taq kit (TaKaRa, Kusatsu, Japan) on the QuantStudio 6 Flex qPCR System (Applied Biosystems, Foster City, CA, USA). The $2^{-\Delta\Delta CT}$ method was used to determine relative gene expression [28]. *V. vinifera EF1γ* (CB977561), *A. thaliana UBQ5* (NM_116090.3) and *APT1* (NM_102509.4) were used as the internal reference genes. The primer sequences are listed in Table S1.

## 3. Results

### 3.1. VaCRK2 Enhances Resistance to B. cinerea in A. thaliana

Four cysteine-rich receptor-like kinase (CRK) genes were identified in *V. amurensis* genome and named *VaCRK1–VaCRK4* according to their position in the genome. The protein encoded by *VaCRK* genes comprises 200 to 420 amino acids in length. The predicted molecular weight is 48 to 66 KDa, and the isoelectric point ranges from 3.7 to 5.2 (Table S2). The four *VaCRK* genes were transformed into *A. thaliana* wildtype Col-0, and at least three overexpression lines were obtained following PCR and RT-PCR analyses. The disease assay results showed that the lesion area and fungal biomass of *VaCRK2* overexpression lines were 56% and 40% lower than those of the wildtype Col-0, respectively, indicating that the *VaCRK2* overexpression lines were more resistant to *B. cinerea* than the wildtype Col-0 (Figure 1). Notably, transgenic *A. thaliana* lines overexpressing the other three *VaCRK* genes *VaCRK1*, *VaCRK3* and *VaCRK4* showed no significant improvement in resistance to *B. cinerea* compared with the wildtype Col-0 (Figure S1).

### 3.2. VaCRK2 Localizes to Plasma Membrane

Conserved domain analysis of *VaCRK2* peptide sequence using SMART identified a typical *CRK* family protein with signal peptide and TM domain, which were confirmed by SignalP5.0 (1–25 aa) and TMHMM2.0 (266–288 aa), respectively (Figure 2A). To further understand the function of *VaCRK2*, the subcellular localization was verified by transient expression of *VaCRK2* and green fluorescent fusion protein (*VaCRK2-GFP*) in tobacco leaf epidermal cells. According to the results, the fluorescent signal of *VaCRK2-GFP* fusion protein was clearly localized to the cell membrane, in contrast to the fluorescence signal of GFP proteins, which was prevalent throughout the foliar cells in tobacco, indicating that VaCRK2 was localized on the cell membrane (Figure 2B).

### 3.3. VaCRK2 Positively Responds to B. cinerea and MeJA Treatment

To determine the relationship between *VaCRK2* expression and *B. cinerea* resistance, we compared the expression patterns of *VaCRK2* between the resistant and susceptible grape cultivars following *B. cinerea* inoculation. *VaCRK2* expression was significantly upregulated in resistant cv. Shuangyou from 2 to 24 h post-inoculation, after which the expression

was reduced (Figure 3A). In contrast, no significant change in *VaCRK2* expression was observed in susceptible cv. Red Globe during the entire experimental period, indicating that *VaCRK2* may mediate *B. cinerea* resistance in grapes. To explore *VaCRK2*-related signaling pathways, we analyzed the expression patterns of *VaCRK2* after MeJA, SA, ETH and ABA treatments. *VaCRK2* was significantly upregulated in resistant cv. Shuangyou after treatment with methyl MeJA within 120 h (Figure 3B) and less affected by treatment with ET (Figure 3D), but it was downregulated by treatment with SA or ABA at several points (Figure 3C,E), which suggested that the *VaCRK2* response of the resistant cultivar to *B. cinerea* is mediated by JA signaling pathway. Overall, these findings show that *B. cinerea* infection and exogenous MeJA induce *VaCRK2* expression in gray mold-resistant cv. Shuangyou.

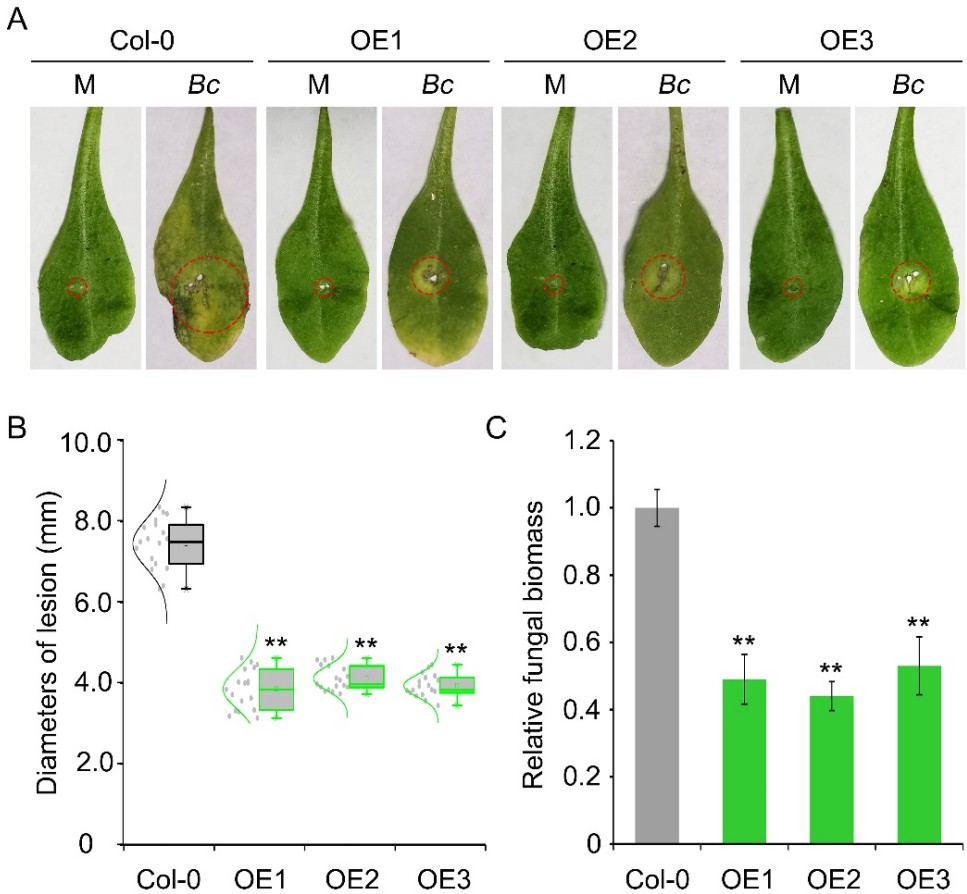

**Figure 1.** *VaCRK2* mediates resistance to *Botrytis cinerea* in *Arabidopsis thaliana*. (**A**) Phenotype of *A. thaliana* wildtype Col-0 and three *VaCRK2* overexpression lines (OE1, OE2 and OE3) after inoculation with *B. cinerea*. (**B**) Relative disease index of *A. thaliana* wildtype Col-0 and three *VaCRK2* OE lines. (**C**) RT-qPCR analysis of relative fungal biomass in *A. thaliana* wildtype Col-0 and three *VaCRK2* OE lines. Error bars represent the standard error of the mean of three independent replicates, asterisks (**) indicate significant difference at $p < 0.01$ (Student's *t*-tests). Mock (M), *Botrytis cinerea* (*Bc*).

### 3.4. VaCRK2 Mediates B. cinerea Resistance Via the JA Signaling Pathway

Jasmonic acid (JA) signaling pathway plays a vital role in plant disease resistance, especially against saprophytic fungal diseases [29]. Therefore, in order to verify whether *VaCRK2* mediates resistance to *B. cinerea* through JA signaling pathway, we firstly detected the expression pattern of *AtCRK2*, a homolog of *VaCRK2* in *A. thaliana*, and the results showed that the expression of *AtCRK2* gene was not induced in *A. thaliana* wildtype Col-0 inoculated with *B. cinerea* (Figure S2). Secondly, we overexpressed *VaCRK2* gene in JA-deficient *A. thaliana* mutant *jar1* (SALK_030821); compared with the mutant *jar1*, the overexpressed line could not improve the resistance to *B. cinerea* (Figure S3). Thirdly, we

examined the expression patterns of genes related to JA signaling pathway in *A. thaliana* wildtype Col-0 and *VaCRK2* overexpression lines after inoculation with *B. cinerea*; the results showed that 11 JA signaling pathway-related genes, including *AOC1*, *AOS*, *OPR3*, *LOX2*, *ACX*, *COI1*, *VSP2*, *ORA59*, *JAR1* and *MYC2*, were upregulated in both wildtype Col-0 and *VaCRK2* overexpression lines but significantly more upregulated in *VaCRK2* overexpression lines than in the wildtype Col-0 (Figure 4, Figures S4 and S5). Altogether, these results indicate that *VaCRK2* interacts with the JA signaling pathway to mediate *B. cinerea* resistance in *A. thaliana*.

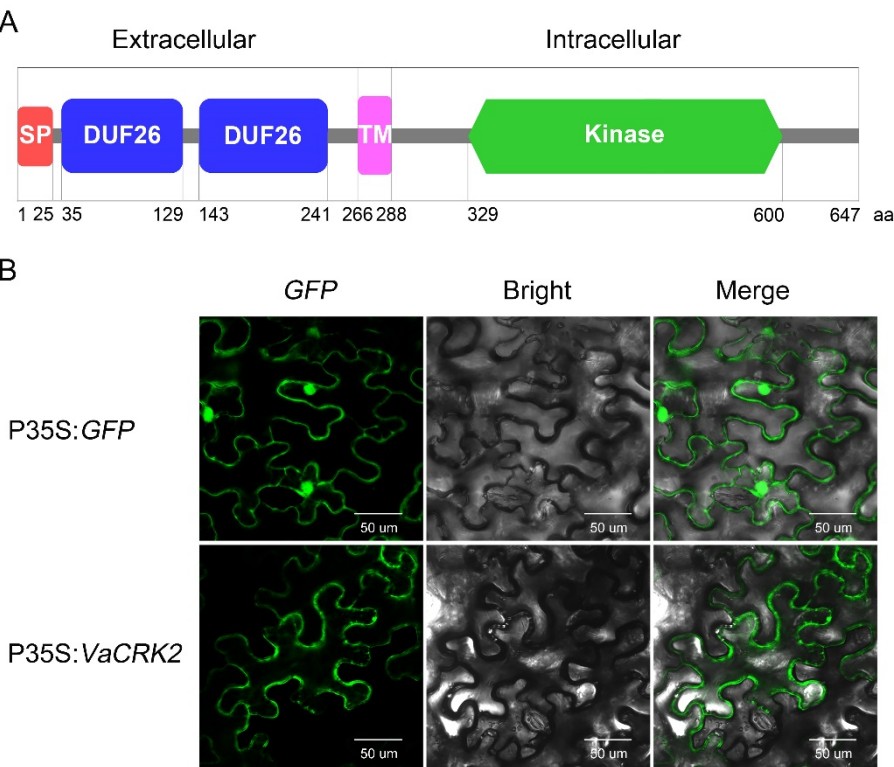

**Figure 2.** VaCRK2 protein structure and its subcellular localization. (**A**) Schematic representation of VaCRK2 protein structure. A signal peptide (SP) from 1 to 25 amino acid (aa), two Stress-antifung (DUF26) domains from 35 to 129 aa and 143 to 241 aa, a transmembrane (TM) domain from 266 to 288 aa and a kinase domain from 347 to 596 aa. (**B**) Subcellular localization of VaCRK2 protein. p35S:*VaCRK2* and p35S:*GFP* vectors were introduced into tobacco epidermal cells by *Agrobacterium* infiltration.

### 3.5. VaCRK2 Overexpression Affects PR Gene Expression and ROS Accumulation

Pathogenesis-related (*PR*) gene expression and ROS accumulation in *A. thaliana* wildtype Col-0 and *VaCRK2* overexpression lines were evaluated after *B. cinerea* inoculation to further verify that *VaCRK2* facilities defense response against *B. cinerea*. The expression levels of JA signaling pathway defense-related genes (*PDF1.2*, *PR3* and *PR4*) were significantly higher in the *VaCRK2* overexpression lines than in the wildtype Col-0 after inoculation with *B. cinerea* (Figure 5A, Figures S6 and S7). Furthermore, ROS accumulation in the leaves of *VaCRK2* overexpression lines was significantly higher than in wildtype Col-0 relative to sterile water treatment (Figure 5B,C). Peroxidase and catalase activity detection results showed that there was no significant difference in peroxidase and catalase activities between wildtype Col-0 and overexpression line of *A. thaliana* under control conditions. After inoculation with *B. cinerea*, the peroxidase and catalase activities in wildtype Col-0 and overexpression line of *A. thaliana* were increased, and the overexpression line showed significantly higher changes than wildtype Col-0 (Figure S8). Altogether, these

results suggest that *VaCRK2* mediates resistance to *B. cinerea* by upregulating *PR* genes and inducing oxidative burst in *A. thaliana*.

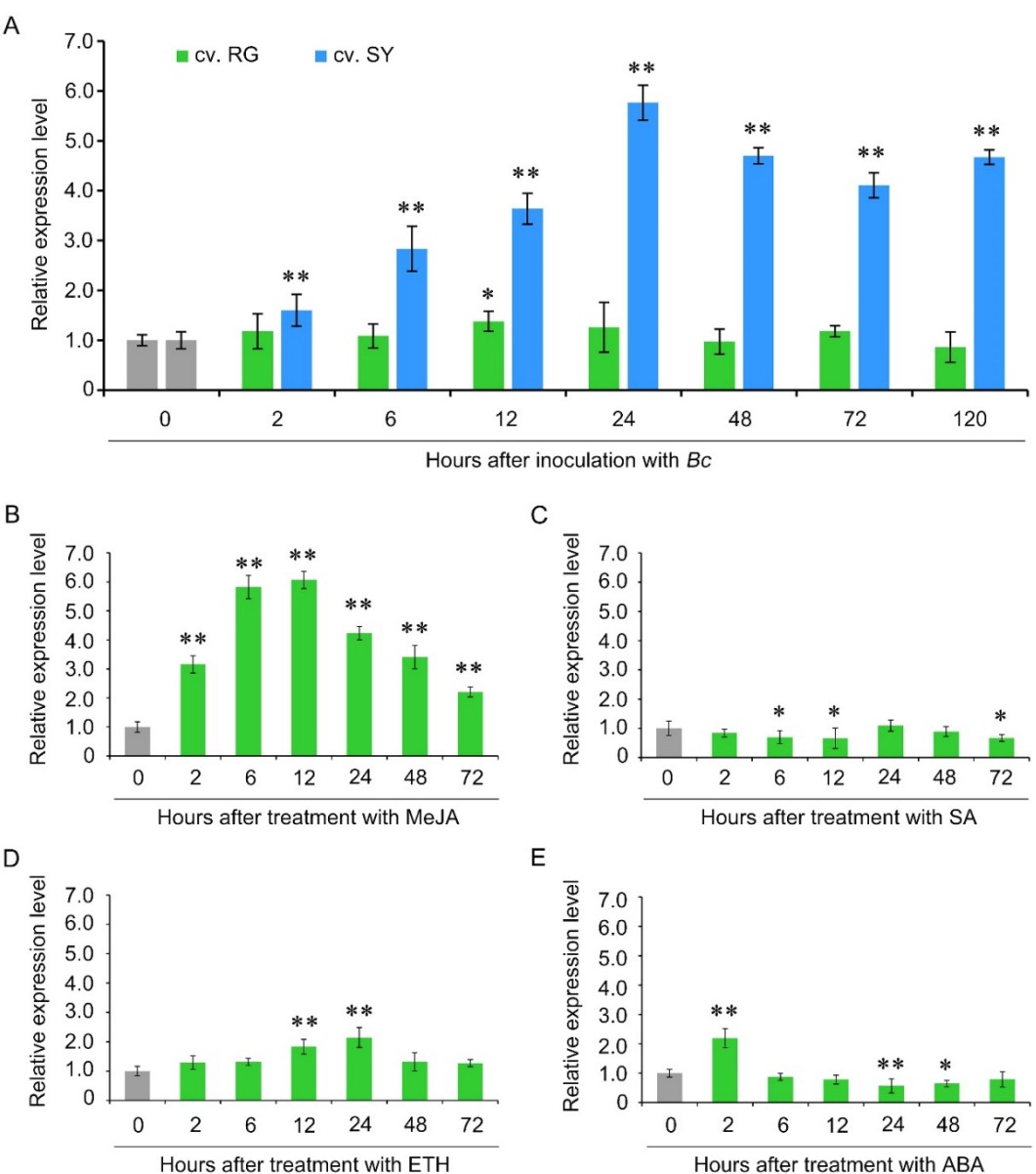

**Figure 3.** Expression patterns of *VaCRK2* under different treatments. (**A**) Expression analysis of *VaCRK2* in resistant *Vitis amurensis* cv. Shuangyou (SY) and susceptible *V. vinifera* cv. Red Globe (RG) after inoculation with *B. cinerea*. Expression analysis of *VaCRK2* in cv. Shuangyou in response to (**B**) methyl jasmonic acid (MeJA), (**C**) salicylic acid (SA), (**D**) ethephon (ETH) and (**E**) abscisic acid (ABA). Error bars represent the standard error of the mean of three independent replicates, asterisks (*) and (**) indicate significant difference at $p < 0.05$ and $p < 0.01$, respectively (Student's *t*-tests). *Botrytis cinerea* (*Bc*).

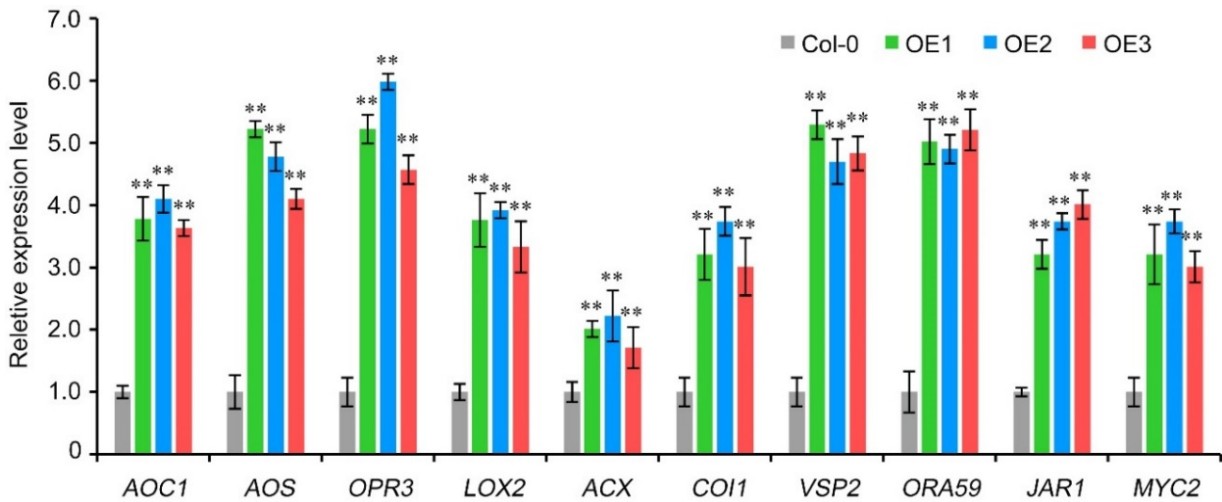

**Figure 4.** *VaCRK2* regulates the jasmonic acid signaling pathway-related genes to confer resistance to *Botrytis cinerea* in *A. thaliana.* Wildtype (Col-0) and *VaCRK2* overexpression lines (OE1, OE2 and OE3) were inoculated with *B. cinerea*, and leaf samples were collected for RT-qPCR analysis 24 h post-inoculation. Error bars represent the standard error of three independent replicates, asterisks (**) indicate significant difference at $p < 0.01$ (Student's *t*-tests).

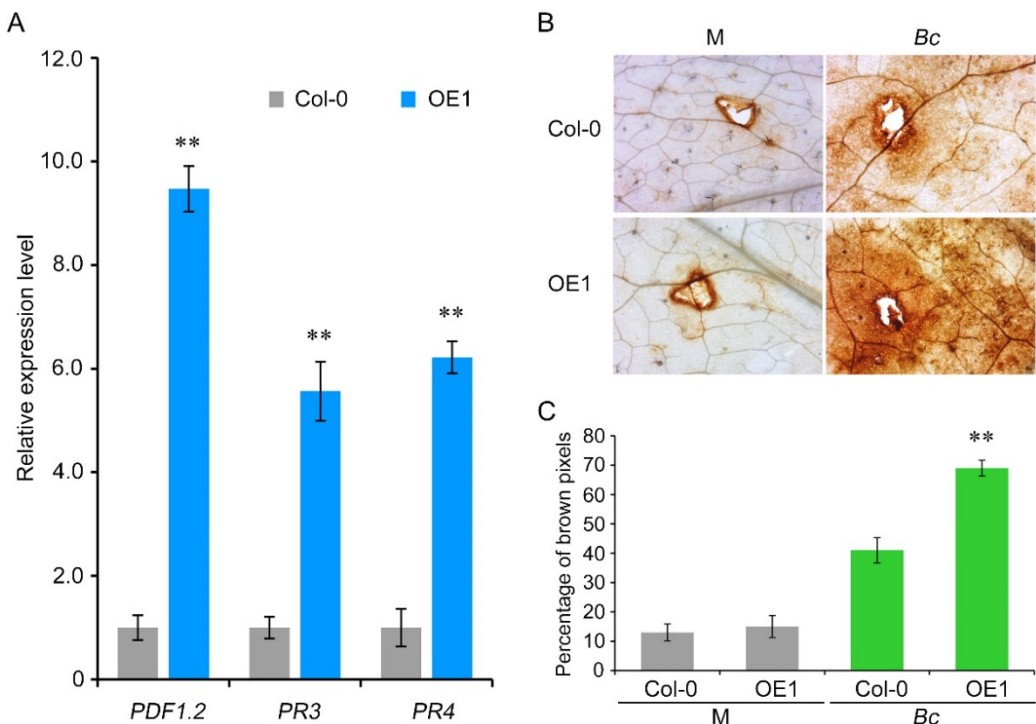

**Figure 5.** *VaCRK2* mediates *Botrytis cinerea* resistance in *Arabidopsis thaliana* by activating pathogenesis-related (PR) genes and oxidative burst. (**A**) Expression analysis of JA signaling pathway PR genes (*PDF1.2*, *PR3* and *PR4*) in wildtype Col-0 and *VaCRK2* overexpression line 1 (OE1) after inoculation with *B. cinerea*. (**B**) Phenotype of wildtype Col-0 and *VaCRK2* OE1 after inoculation with *B. cinerea* followed by diaminobenzidine (DAB) staining. (**C**) The browning intensity of wildtype Col-0 and *VaCRK2* OE1 leaves after inoculation with *B. cinerea* followed by DAB staining. Error bars represent the standard error of the mean of three independent biological replicates, and asterisks (**) indicate significant difference at $p < 0.01$ (Student's *t*-tests). Mock (M), *Botrytis cinerea* (*Bc*).

## 4. Discussion

Cysteine-rich receptor-like kinases (CRKs) have been shown to play an essential role in mediating disease resistance in several plants, including *Arabidopsis* [10,11,30], rice [13], barley [14], wheat [31] and *Medicago truncatula* [32]. However, *CRK* genes and their function

in disease resistance have not been demonstrated in grapes. This study characterized *CRK* genes in *V. amurensis* (*VaCRK*) and evaluated their role in mediating *B. cinerea* resistance. The *VaCRK* genes identified in this study have a signal peptide and two DUF26 motifs at the N-terminus, a TM domain in the middle and a kinase domain at the C-terminus, which are typical domains of a CRK gene. Based on the CRK gene identification criteria, five candidate genes were identified in *V. amurensis*. However, *VAG0113336.1* and *VAG0128137.1* genes were the same and were combined into one gene, *VaCRK3* (Table S2). Among the four *CRK* genes identified in *V. amurensis*, *VaCRK2* was the only one shown to confer resistance to *B. cinerea* effectively. *VaCRK2* is located on the cell membrane and significantly increased the resistance of *A. thaliana* overexpression lines to *B. cinerea* (Figures 1 and 2). Studies have shown that ETI promotes PTI by inducing the expression of PTI signal components, while PTI ensures the normal resistance function of ETI through diverse signals [33]. Enhanced expression of immune system-related genes, especially PTI genes, is crucial in mediating plant disease resistance [34]. *VaCRK2* expression was significantly increased in resistant cv. Shuangyou at several time points after *B. cinerea* inoculation, but not in susceptible cv. Red Globe (Figure 3A), suggesting that *VaCRK2* potentially mediates *B. cinerea* resistance and its expression is cultivar-dependent. At the same time, it is speculated that this result may be because the host increased the transcription level of *VaCRK2* gene through immune regulation after sensing the signal of *B. cinerea*, especially in the early stage, to resist the infection of *B. cinerea* [16,17]. Perhaps because of this, different grape varieties have resistant and susceptible phenotypes to pathogens, which is consistent with the disease resistance mediated by CRKs in other plants [6].

JA is an essential signaling molecule in plants and facilitates various physiological reactions, including seed germination, damage response and biological stress response [29]. In this study, exogenous MeJA application induced the expression of *VaCRK2* gene (Figure 3B), indicating that *B. cinerea* resistance mediated by *VaCRK2* may be related to the JA signaling pathway. *VaCRK2* gene expression was upregulated at 12 and 24 h after ETH treatment and downregulated at 6, 12 and 72 h after SA treatment (Figure 3C,D), which may be because ETH and JA act synergistically, and SA has negative feedback regulation on JA [29]. The JA signaling pathway activates diverse defense-related proteins and secondary metabolites in response to necrotrophic fungal invasion [29]. *B. cinerea*, the causative agent of gray mold disease, is a typical necrotrophic fungus infecting several plant species. Several genes related to JA signaling pathway have been cloned from different plants. These include JA self-synthesis genes, such as *LOX2*, *AOS* and *OPR3*, and defense-related genes, such as *PDF1.2*, *PR3* and *PR4* [35]. In this study, several genes related to JA signaling pathway were upregulated in *A. thaliana VaCRK2* overexpression lines compared with *A. thaliana* wildtype Col-0 after *B. cinerea* inoculation (Figure 4, Figures S2 and S8). The JA signaling pathway has been associated with gray mold resistance in different plants, including *A. thaliana*, cucumber and tomato [36–38]. For example, *A. thaliana phyB* and *SDG8* genes participate in the resistance defense against *B. cinerea* by regulating JA biosynthesis [39,40]. In addition, *SlMYB75* plays an active role in the resistance of tomato to *B. cinerea* by regulating the JA signaling pathway [38]. Therefore, we speculated that the JA signaling pathway may be essential in *VaCRK2*-mediated resistance against *B. cinerea*.

*PR* gene expression is upregulated in many plants after inoculation with phytopathogens, including fungi, bacteria or viruses, to attack pathogens and inhibit pathogen infection [41]. Oxidative burst plays a vital role in plant disease resistance by directly neutralizing pathogens and amplifying signals of other immune events [4]. Notably, ROS accumulation can directly kill the invading pathogens and activate cell wall cross-linking to restrict further pathogen invasion [42]. Plant resistance proteins, including CRK proteins, recognize specific pathogenic effectors and initiate various defense responses, such as *PR* gene overexpression and ROS accumulation [43,44]. For instance, overexpression of *CRK5*, *CRK6*, *CRK36* and *CRK45* in *A. thaliana* significantly enhances the resistance to *Pseudomonas syringae* pv. *tomato* DC3000 by upregulating PR genes and enhancing ROS accumulation [9,30,45]. *CRK6* and *CRK7* genes play an essential function in extracellular ROS signal

transduction in *A. thaliana* [46]. Various defense responses, including hormone induction, *PR* gene overexpression and ROS accumulation, are key to grape resistance against gray mold disease [16,47]. In this study, JA signaling pathway defense-related PR gene expression and ROS accumulation in *A. thaliana VaCRK2* overexpression lines were significantly higher than in wildtype Col-0 after *B. cinerea* inoculation (Figure 5, Figures S3 and S6), indicating that *VaCRK2* mediates gray mold resistance through JA signaling pathway defense-related PR gene expression and ROS accumulation.

## 5. Conclusions

This study identified and characterized *CRK* genes in *V. amurensis* and validated their role in *B. cinerea* resistance. The overexpression of *VaCRK2* significantly increased the resistance to *B. cinerea* in *A. thaliana,* suggesting that *VaCRK2* may play a crucial role in gray mold resistance. Furthermore, *VaCRK2* expression significantly induced the JA signaling pathway-related genes, PR gene expression and ROS accumulation after *B. cinerea* inoculation, indicating that *VaCRK2* mediates *B. cinerea* resistance by interacting with the JA signaling pathway and activating oxidative burst. This study provides new insights into the role of *CRKs* in plant disease resistance and will inform future research on the molecular breeding of grapes for resistance against gray mold disease.

**Supplementary Materials:** The following are available online at https://www.mdpi.com/article/10.3390/agronomy11081672/s1, Figure S1. Performance of *A. thaliana* wildtype Col-0 and *VaCRK1*, *VaCRK3*, and *VaCRK4* overexpression lines following inoculation with *B. cinerea*. Figure S2. Expression analysis of *AtCRK2* in *A. thaliana* wildtype Col-0 after inoculation with *B. cinerea*. Figure S3. *VaCRK2* cannot mediates resistance to *B. cinerea* in *A. thaliana* mutant *jar1*. Figure S4. JA signaling pathway-related genes expression pattern in *A. thaliana* wildtype Col-0 and *VaCRK2* overexpression lines (OE1, OE2, and OE3) after inoculation with *B. cinerea*. Figure S5. JA signaling pathway-related genes expression pattern in *A. thaliana* wildtype Col-0 and *VaCRK2* overexpression line1 (OE1) before and after inoculation with *B. cinerea*. Figure S6. Expression analysis of JA signaling pathway *PR* genes (*PDF1.2*, *PR3*, and *PR4*) in *A. thaliana* wildtype Col-0 and *VaCRK2* overexpression line 1 (OE1) after inoculation with *B. cinerea*. Figure S7. Expression analysis of JA signaling pathway *PR* genes (*PDF1.2*, *PR3*, and *PR4*) in wildtype Col-0 and *VaCRK2* overexpression lines (OE1, OE2, and OE3) after inoculation with *B. cinerea*. Figure S8. Detection of the peroxidase (POD) and catalase (CAT) enzyme activity in *A. thaliana* wildtype Col-0 and *VaCRK2* overexpression line 1 (OE1) after inoculation with *B. cinerea*. Table S1. Information of the PCR primers used in this study. Table S2. Information of candidate *VaCRK* genes in *Vitis amurensis*.

**Author Contributions:** Writing—original draft preparation, T.L.; methodology, H.G.; data curation, X.T.; conceptualization, D.G. All authors have read and agreed to the published version of the manuscript.

**Funding:** This research was funded by the National Natural Science Foundation of China (31901866).

**Institutional Review Board Statement:** Not applicable.

**Informed Consent Statement:** Not applicble.

**Data Availability Statement:** Not applicable.

**Conflicts of Interest:** The authors declare no conflict of interest.

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
