# Peer review of "VaCRK2 Mediates Gray Mold Resistance in Vitis amurensis by Activating the Jasmonate Signaling Pathway"

_agronomy, doi:10.3390/agronomy11081672_

Round 1

Reviewer 1 Report

This research article entitled "VaCRK2 Mediates Gray Mold Resistance in Vitis amurensis by

Activating the Jasmonate Signaling Pathway" aimed to highlight the impact of an in-silico identified Cystine-like receptor kinase from the grape plant, Vitis amurensis, in boosting the defenses against the necrotrophic plant fungus, Botrytis cinerea in the genetically transformed Arabidopsis thaliana line.

The article is well written (except for minor mistakes) and the article structure is clear.

However, the experimental design and description/discussion lack crucial information:

Minor:

  1. In the introduction line 72, the authors stated that the host range of B.  cinerea is > 200 supported with an outdated reference (2007). Please revise these articles for the updated information.

https://doi.org/10.1186/s12864-019-5580-x                   and   

         https://doi.org/10.1007/978-3-319-23371-0_20"

  1. In line 188, please specify the bioinformatics analysis and tool(s) by which you obtained these results.
  2. In figure 2B, Blight field should be “Bright” field
  3. In the same figure, the scalebar on 2B is missing
  4. In lines 245 and 292, I think the authors meant to say PR3 instead of VSP3.
  5. In line 288, I think the authors meant to use the word “agent” not “urgent” 

Major:

  1. In the material and methods, the primer sequences used in this study is not provided in the main text nor in the supplementary material, except for the housekeeping gene UBQ5, where the reference is cited. Please provide all oligonucleotides used in the study.
  2. The provided links for the databases used in this study for the in-silico analysis (Identification of the VaCRKs) are not accessible (page not found on 21st July 2021)
  3. qRT-PCR was performed and normalized based on one housekeeping gene (UBQ5), although using at least two is recommended.
  4. in the M&M line 130, the authors refer to a fragment of the VaCRK2 during the transformation of the Arabidopsis plants. First, from the study I understood that the transformation of Arabidopsis involved the whole gene, not just a fragment (e.g., line 132). Moreover, no information about this fragment is indicated (size, partition, ...). I, however, think this is a typo and the authors meant the whole gene, am I correct? Please verify and/or correct this information.
  5. in the results section:
    1. since Arabidopsis already has the CRK2 gene, I think it is important to show the sequence similarity or percentage identity between the VaCRK2 and the Also, is AtCRK2 expressed under the tested conditions?
    2. Regarding the previous point, I think the experimental design would have been stronger and informative if the authors used AtCRK2-deficient line (T-DNA or CRISPR line) as a background for transformation with VaCRK2. This way, the effect can be unambiguously assigned due to overexpression of the VaCRK2 gene and not due to other factors, such as AtCRK2 activity.
    3. In figure 1B, it is It is important to see all the data points since it shows where most of the observations lie on the range of the replicates.
    4. In line 212, the authors claim that the VaCRK2 expression is not influenced by the treatment except for MeJA. However, according to figure 3B-E, these treatments still had significant effects on the expression level of the VaCRK2, especially the treatments after 12 and 24 h of ETH and 2 h of ABA. In addition, there is a slight yet reduction significant upon SA treatment. The effect is not as strong as with MeJA, but it is still worth mentioning and discussing this observation. This could be a feedback effect upon elicitation of the JA signaling pathway
    5. Figure 5A shows the expression of the three genes of only one over-expressing line (OE1). I wonder if the other lines behaved the same way or were there a significant difference among the three VaCRK2 lines?
    6. In figure 5B-C and the ROS accumulation study, could these brown pixels be due to the necrosis lesion caused by the pathogen ( cinerea) as it is one important feature of the successful pathogenesis of this fungus. It would be useful to confirm these results by measuring the peroxidase and catalase enzyme activity as an indication of accumulated ROS upon treatment with the pathogen.
  6. In the discussion section:
    1. The discussion in general is poor and needs to be significantly enriched. So far it is just reciting the results without critical assessments of the data. For instance, in lines 279-281, the authors stated again the results were the VaCRK2 gene expression is expressed higher in the resistant grape strain compared to the susceptible one. In this case, as a reader, I would like to know why this expression is lower in the susceptible cultivar, especially since the gene exists as shown in the figure?
    2. In lines 299-300 where the authors speculated the involvement of the JA signaling pathway during the expression of VaCRK2. To support this statement, I wonder if the authors could inspect the plant defense response against Botrytis using JA-deficient Arabidopsis line (e.g., cpm2 mutant) overexpressing VaCRK2.

Author Response

Response to Reviewer 1 Comments

Minor:

Point 1: In the introduction line 72, the authors stated that the host range of B. cinerea is > 200 supported with an outdated reference (2007). Please revise these articles for the updated information. https://doi.org/10.1186/s12864-019-5580-x and https://doi.org/10.1007/978-3-319-23371-0_20

Thank you for the suggestion. We have now incorporated the same in the revised version of the manuscript.

Point 2: In line 188, please specify the bioinformatics analysis and tool(s) by which you obtained these results.

Thank you for the suggestion. We have added the bioinformatics analysis website in section 3.2 of our revised manuscript.

Point 3: In figure 2B, Blight field should be “Bright” field.

Thank you for the suggestion. We have revised figure 2B.

Point 4: In the same figure, the scalebar on 2B is missing.

Thank you for the suggestion. We have added the scalebar on figure 2B.

Point 5: In lines 245 and 292, I think the authors meant to say PR3 instead of VSP3.

Thank you for pointing it out. We have revised in our revised manuscript.

Point 6: In line 288, I think the authors meant to use the word “agent” not “urgent”.

Thank you for the careful revision. We have revised in our revised manuscript.

Major:

Point 1: In the material and methods, the primer sequences used in this study is not provided in the main text nor in the supplementary material, except for the housekeeping gene UBQ5, where the reference is cited. Please provide all oligonucleotides used in the study.

Thank you for the suggestion. We have added the primer sequences used in this study in Table S2.

Point 2: The provided links for the databases used in this study for the in-silico analysis (Identification of the VaCRKs) are not accessible (page not found on 21st July 2021).

Thank you for your reminder. We have reset the website involved in the “Identification of the VaCRKs” section.

Point 3: qRT-PCR was performed and normalized based on one housekeeping gene (UBQ5), although using at least two is recommended.

Thank you for the suggestion. We used APT1 as another housekeeping gene to detect the gene expression involved in the manuscript (Hocq et al., 2020). The expression level is consistent with the trend of taking UBQ5 as the housekeeping gene, and the results are shown in Figure S2 and S3.

Hocq L., Guinand S., Habrylo O., Voxeur A., Tabi W., Safran J., et al. (2020) The exogenous application of AtPGLR, an endo-polygalacturonase, triggers pollen tube burst and repair. Plant J., 103, 617-633.

Point 4: In the M&M line 130, the authors refer to a fragment of the VaCRK2 during the transformation of the Arabidopsis plants. First, from the study I understood that the transformation of Arabidopsis involved the whole gene, not just a fragment (e.g., line 132). Moreover, no information about this fragment is indicated (size, partition, ...). I, however, think this is a typo and the authors meant the whole gene, am I correct? Please verify and/or correct this information.

Thank you for the suggestion. You're right. This refers to the whole gene rather than gene fragment, which have been modified in our revised manuscript.

Point 5: In the results section:

5.1 Since Arabidopsis already has the CRK2 gene, I think it is important to show the sequence similarity or percentage identity between the VaCRK2 and the Also, is AtCRK2 expressed under the tested conditions?

Thank you for the suggestion. There was 18.65% sequence similarity between VaCRK2 and AtCRK2. The results showed that the expression of AtCRK2 gene was not induced in Arabidopsis wild type inoculated with Botrytis cinerea (Figure S4). Therefore, we speculated that the AtCRK2 gene is not involved in the response to B. cinerea inoculation.

5.2 Regarding the previous point, I think the experimental design would have been stronger and informative if the authors used AtCRK2-deficient line (T-DNA or CRISPR line) as a background for transformation with VaCRK2. This way, the effect can be unambiguously assigned due to overexpression of the VaCRK2 gene and not due to other factors, such as AtCRK2 activity.

Thank you for the suggestion. It has been proved that AtCRK2 gene can mediate the resistance of Pseudomonas syringae pv tomato DC3000 (Kimura et al., 2020), but so far there is no evidence that AtCRK2 gene can mediate fungal resistance. At the same time, AtCRK2 gene cannot respond to Botrytis cinerea inoculation, we believe that AtCRK2 gene has no significant effect on proving the B. cinerea resistance function of VaCRK2 in Arabidopsis thaliana wildtype Col-0.

Kimura S., Hunter K., Vaahtera L., Tran H.C., Citterico M., Vaattovaara A., et al. (2020) CRK2 and C-terminal phosphorylation of NADPH oxidase RBOHD regulate reactive oxygen species production in Arabidopsis. The plant cell, 32, 1063–1080.

5.3 In figure 1B, it is It is important to see all the data points since it shows where most of the observations lie on the range of the replicates.

Thank you for the suggestion. We have modified figure 1B to show all data points.

5.4 In line 212, the authors claim that the VaCRK2 expression is not influenced by the treatment except for MeJA. However, according to figure 3B-E, these treatments still had significant effects on the expression level of the VaCRK2, especially the treatments after 12 and 24 h of ETH and 2 h of ABA. In addition, there is a slight yet reduction significant upon SA treatment. The effect is not as strong as with MeJA, but it is still worth mentioning and discussing this observation. This could be a feedback effect upon elicitation of the JA signaling pathway.

Thank you for the suggestion. We added a description of this observation in the “Results 3.3” and “Discussion” sections of the revised manuscript respectively.

5.5 Figure 5A shows the expression of the three genes of only one over-expressing line (OE1). I wonder if the other lines behaved the same way or were there a significant difference among the three VaCRK2 lines?

Thank you for the suggestion. We tested the expression of the PR genes in three transgenic lines, and the results showed that the expression pattern of the PR gene in the OE1 line was consistent with OE2 and OE3 lines, and there was no significant difference between the three transgenic lines (Figure S5).

5.6 In figure 5B-C and the ROS accumulation study, could these brown pixels be due to the necrosis lesion caused by the pathogen (B. cinerea) as it is one important feature of the successful pathogenesis of this fungus. It would be useful to confirm these results by measuring the peroxidase and catalase enzyme activity as an indication of accumulated ROS upon treatment with the pathogen.

Thank you for the suggestion. We added peroxidase and catalase activity detection, as shown in Figure S6. The results showed that there was no significant difference in peroxidase and catalase activities between wildtype and overexpression (OE) line of Arabidopsis thaliana under control conditions. After inoculation with Botrytis cinerea, the peroxidase and catalase activities in wildtype and OE line of A. thaliana were increased, and the OE line showed significantly higher changes than this of wildtype.

Point 6: In the discussion section:

6.1 The discussion in general is poor and needs to be significantly enriched. So far it is just reciting the results without critical assessments of the data. For instance, in lines 279-281, the authors stated again the results were the VaCRK2 gene expression is expressed higher in the resistant grape strain compared to the susceptible one. In this case, as a reader, I would like to know why this expression is lower in the susceptible cultivar, especially since the gene exists as shown in the figure?

Thank you for the suggestion. We have supplemented and improved the “Discussion” section in our revised manuscript, especially lines 279-281.

6.2 In lines 299-300 where the authors speculated the involvement of the JA signaling pathway during the expression of VaCRK2. To support this statement, I wonder if the authors could inspect the plant defense response against B. cinerea using JA-deficient Arabidopsis line (e.g., cpm2 mutant) overexpressing VaCRK2.

Thank you for the suggestion. We have carried out this work. The results show that overexpression of VaCRK2 in JA deficient Arabidopsis mutant jar1 (SALK_030821) cannot improve the resistance to Botrytis cinerea, indicating that JA signaling pathway is necessary for VaCRK2 gene mediated resistance (Figure S7).

Reviewer 2 Report

In the work presented in the manuscript “VaCRK2 Mediates Gray Mold Resistance in Vitis amurensis by Activating the Jasmonate Signaling Pathway” the authors identified four Cystein-rich receptor-like kinases (CRKs) in Vitis amurensis. CRK genes are involved in the resistance against gray mold caused by Botritis cinerea in grape. The authors investigated the function of CRKs ectopically expressing the genes in A.thaliana showing that VaCRK2 can confer resistance to B.cinerea in A.thaliana. In addition, gene expression analysis carried out in VaCRK2-overexpression lines of A.thaliana showed that there is up-regulation of genes involved in jasmonic acid signaling pathway after B.cinerea inoculation and up-regulation of pathogenesis-related genes as well as accumulation of reactive oxygen species.

The manuscript is well written and the results could be very interesting for the scientific community.

One minor issues that can be easily addressed:

  • In paragraph 3.2 “According to the results, the fluorescent signal of VaCRK2-GFP fusion protein was only observed on the cell membrane, while GFP fluorescent signal alone was observed in tobacco leaf epidermal cells, indicating that VaCRK2 is located on the cell membrane (Figure 2B)” this sentence should be improved since the difference in localization is not well described. What is the meaning of “observed in tobacco leaf epidermal cells”?
  • In materials and methods under paragraph 2.2 a reference for the vector used should be reported.

However the main concern that I have is the following:

The authors show that VaCRK2 is up-regulated during B.cinerea infection and based on their results it should mediate resistance interacting with JA signaling pathway and promoting PR Gene expression and ROS accumulation. If so, then in A.thaliana VaCRK2 overexpression lines the expression of the genes shown to be differentially regulated after B.cinerea inoculation could be always affected even without the inoculation with the pathogen. To assess this point, the gene expression analysis should be performed also in non-inoculated transgenic lines for the comparison before and after inoculation with B.cinerea. From the results presented in the work and the materials and methods, I understand that the relative expression was calculated against the Col-0 values (after normalization against the reference gene). If this is the case, then the effect of B.cinerea on the gene expression in the transgenic lines is not evaluated correctly. Consequently, the main text would considerably change and further data should be added to assess whether the effect on gene expression is due to the overexpression of VaCRK2 or if the presence of B.cinerea is required to observe the described behavior.

If the authors already did the comparison against the non-inoculated transgenic lines then this can be considered a major issue and should be explained in materials and methods and the main text improved accordingly.

Author Response

Response to Reviewer 2 Comments

Minor issues:

Point 1: In paragraph 3.2 “According to the results, the fluorescent signal of VaCRK2-GFP fusion protein was only observed on the cell membrane, while GFP fluorescent signal alone was observed in tobacco leaf epidermal cells, indicating that VaCRK2 is located on the cell membrane (Figure 2B)” this sentence should be improved since the difference in localization is not well described. What is the meaning of “observed in tobacco leaf epidermal cells”?

Thank you for the suggestion. We have improved the description of this part in our revised manuscript.

Point 2: In materials and methods under paragraph 2.2 a reference for the vector used should be reported.

Thank you for the suggestion. We have cited the references of the vector used in our revised manuscript.

Main concern:

The authors show that VaCRK2 is up-regulated during B. cinerea infection and based on their results it should mediate resistance interacting with JA signaling pathway and promoting PR gene expression and ROS accumulation. If so, then in A. thaliana VaCRK2 overexpression lines the expression of the genes shown to be differentially regulated after B. cinerea inoculation could be always affected even without the inoculation with the pathogen. To assess this point, the gene expression analysis should be performed also in non-inoculated transgenic lines for the comparison before and after inoculation with B. cinerea. From the results presented in the work and the materials and methods, I understand that the relative expression was calculated against the Col-0 values (after normalization against the reference gene). If this is the case, then the effect of B. cinerea on the gene expression in the transgenic lines is not evaluated correctly. Consequently, the main text would considerably change and further data should be added to assess whether the effect on gene expression is due to the overexpression of VaCRK2 or if the presence of B. cinerea is required to observe the described behavior.

If the authors already did the comparison against the non-inoculated transgenic lines then this can be considered a major issue and should be explained in materials and methods and the main text improved accordingly.

Thank you for the valuable suggestion. At first, we performed the expression pattern of JA signaling pathway-related genes in Arabidopsis thaliana wildtype Col-0 and VaCRK2 gene overexpression line 1 (OE1) without Botrytis cinerea inoculation, the results showed that the related genes in OE1 line was significantly up-regulated compared with wildtype Col-0. Secondly, we compared the expression pattern of JA signaling pathway-related genes in A. thaliana wildtype Col-0 before and after inoculation with B. cinerea, the results showed that the expression of related genes was significantly up-regulated after inoculation with B. cinerea. Thirdly, we compared the expression pattern of JA signaling pathway-related genes in A. thaliana OE1 line before and after inoculation with B. cinerea, the results showed that the expression of related genes was also significantly up-regulated after inoculation with B. cinerea. We comprehensively show the above experimental results in Figure S8.

The above results show that the overexpression of VaCRK2 gene would significantly up-regulate the expression of JA signaling pathway-related genes. Under the condition of B. cinerea inoculation, the expression of JA signaling pathway-related genes would be up-regulated in both wildtype Col-0 and OE1 line, but it showed more significant up-regulation in OE1 line, resulting in the enhancement of resistance of OE1 line to B. cinerea, it shows that VaCRK2 gene mediates resistance to B. cinerea by activating JA signaling pathway in A. thaliana. In the manuscript, Figure 4 shows the expression pattern of JA signaling pathway-related genes of A. thaliana VaCRK2 gene OE lines (OE1, OE2, OE3) and wildtype Col-0 under the inoculation with B. cinerea, and the result showed that the expression pattern is consistent with that shown in Figure S8. And we have added relevant description in “Results” section of the revised manuscript.

Round 2

Reviewer 1 Report

additional experiments and substantial improvements have been made in the revised version. All my questions/concerns have been answered except one minor point: the word "Blight" in figure 2B still not corrected.

Reviewer 2 Report

The authors of the manuscript “VaCRK2 Mediates Gray Mold Resistance in Vitis amurensis by Activating the Jasmonate Signaling Pathway” resolved the issues highlighted in the first review. The results presented were improved and the authors answered to my main concern explaining how they calculated the relative expression.

However, they added many figures as supplementary data but there is no reference to these data in the main text. Only Table S1, Figure S1 and Figures S8-10 are mentioned in the results part but all the other figures are added to the supplementary files without a reference in the text to link the supplementary supporting data. In addition, the authors performed gene expression analysis with a further gene as a reference (A.thaliana APT1) but this is not reported in materials and methods (only A. thaliana UBQ5 is stated).

The authors explained very well in the response to my review how they proceeded to calculate the relative expression. To make this clear for the readers as well, I suggest adding a short sentence in the material and methods explaining that control plants for the wildtype Col-0 and the transgenic lines where mock inoculated, as they did for the grape experiment. This missing description was generating the misunderstanding in the previous version of the manuscript. (It may be considered obvious, though is not,  that there is a mock control but it should be written in the text).

Last, at line 289 “It is precisely because of this that the grape varieties have different phenotypes that are resistant….” I find the sentence too strong in a context where the arguments ad considerations are presented in a “hypothetical” way and uncertain as one reader can get from the sentences “.. suggesting that VaCRK2 potentially mediates B. cinerea resistance…” ; “..it is speculated that this result may be because…”.

Author Response

Reviewer 2

  1. The authors of the manuscript “VaCRK2 Mediates Gray Mold Resistance in Vitis amurensis by Activating the Jasmonate Signaling Pathway” resolved the issues highlighted in the first review. The results presented were improved and the authors answered to my main concern explaining how they calculated the relative expression. However, they added many figures as supplementary data but there is no reference to these data in the main text. Only Table S1, Figure S1 and Figures S8-10 are mentioned in the results part but all the other figures are added to the supplementary files without a reference in the text to link the supplementary supporting data.

Thank you for the suggestion. We have added the references to link the supplementary supporting data in our revised manuscript.

  1. In addition, the authors performed gene expression analysis with a further gene as a reference (A. thaliana APT1) but this is not reported in materials and methods (only A. thaliana UBQ5 is stated).

Thank you for the suggestion. We have added the report of internal reference gene APT1 of A. thaliana in part 2.7 in our revised manuscript.

  1. The authors explained very well in the response to my review how they proceeded to calculate the relative expression. To make this clear for the readers as well, I suggest adding a short sentence in the material and methods explaining that control plants for the wildtype Col-0 and the transgenic lines where mock inoculated, as they did for the grape experiment. This missing description was generating the misunderstanding in the previous version of the manuscript. (It may be considered obvious, though is not, that there is a mock control but it should be written in the text).

Thank you for the suggestion. We have added the description of control group and experimental group of different A. thaliana plants in part 2.5 and 2.6 in our revised manuscript.

  1. Last, at line 289 “It is precisely because of this that the grape varieties have different phenotypes that are resistant….” I find the sentence too strong in a context where the arguments and considerations are presented in a “hypothetical” way and uncertain as one reader can get from the sentences “.. suggesting that VaCRK2 potentially mediates B. cinerea resistance…” ; “..it is speculated that this result may be because…”.

Thank you for the suggestion. We have improved the expression of the sentence “It is precisely because of this that the grape varieties have different phenotypes that are resistant…” at line 289-291 in our revised manuscript.